# Health seeking behaviors of nurses diagnosed with hypertension and providing health care in resource-constrained setting in a rural part of Northern Ghana: A qualitative study

**Thomas Ankomanin, Kennedy Dodam Konlan**[ID]*, **Kwadwo Ameyaw Korsah**

Department of Adult Health, School of Nursing and Midwifery, University of Ghana, Legon, Greater Accra Region, Ghana

* kennedy.konlan@gmail.com

## Abstract

### Background

Hypertension poses a significant occupational health threat to nurses globally, exacerbated by demanding work environments that can hinder their own health-seeking behaviors. In Ghana, despite a high burden of non-communicable diseases and the existence of employee health and wellbeing programme, evidence on how nurses with hypertension navigate their personal health management remains limited, particularly in high-stress settings like those in rural part of Ghana.

### Aim

We explored the health seeking behaviors of nurses diagnosed with hypertension and providing health care in a resource-constrained setting in a rural part of Northern Ghana.

### Methods

The study employed a qualitative, exploratory-descriptive design. Through purposive sampling, twelve (12) nurses diagnosed with hypertension were recruited from the Kintampo Municipal Hospital's Employee Wellness Clinic. Data was collected via in-depth, semi-structured interviews which were audio-taped and transcribed verbatim. Thematic analysis, guided by Braun and Clarke's framework and supported by NVivo 14.0 software, was used to assist with the data analysis.

### Results

The analysis yielded nine major themes. The participants demonstrated high biomedical knowledge and perceived hypertension as a severe condition, strongly linked to occupational stress. Key barriers to effective management included overwhelming workload and scheduling constraints, financial limitations, a pervasive culture

**Data availability statement:** All relevant data are within the paper and its Supporting Information files.

**Funding:** The author(s) received no specific funding for this work.

**Competing interests:** The authors have declared that no competing interests exist.

**Abbreviations:** BP, blood pressure; CVDs, Cardiovascular Diseases; GHS, Ghana Health Service; HPT, hypertension; NCDs, Non-Communicable Diseases; OPD, Out-Patient Department; SSA, sub-Saharan Africa.

of professional self-reliance and stigma, and a critical lack of institutional support systems. While family and peer support were vital facilitators, they were insufficient to overcome systemic barriers. Nurses exhibited strong self-management practices, but their health-seeking was often reactive rather than preventive.

## Conclusion and recommendations

A significant gap exists between nurses' knowledge of hypertension and their health-seeking actions. This gap is primarily driven by organizational and systemic barriers within the workplace, rather than a lack of individual awareness. To protect this critical workforce, we recommend that nurse managers, hospital management and health policymakers must prioritize the implementation of structured, low-cost workplace wellness programmes. These should include routine screening, flexible scheduling, anti-stigma campaigns to promote health-seeking, and confidential peer support systems to enable nurses to translate their knowledge into consistent self-care practices when diagnosed with hypertension.

## Introduction

Hypertension, often described as the "silent killer," is a leading risk factor for cardiovascular diseases and premature deaths worldwide [1–3]. Globally, an estimated 1.28 billion adults aged 30–79 years are living with hypertension, with two-thirds residing in low- and middle-income countries [1,3]. In Africa, about 46% of adults aged 25 years and above have hypertension [2,4]. Hypertension is a major public health issue in sub-Saharan Africa, driven by risk factors such as physical inactivity, unhealthy diets, alcohol use, obesity, and demographic transitions [1–5]. In Ghana, prevalence rates continue to rise, with fewer than half of those affected being aware of their status and even fewer having their blood pressure under control [6–8]. This situation increases the risk of cardiovascular morbidity and premature death [2,6]. Other national studies in Ghana suggest that nearly one in four adults live with the condition [4–8].

The Ghana Health Service (GHS) launched the Employee Health and Wellbeing Program in 2018 to reduce the burden of non-communicable diseases (NCDs), including hypertension and diabetes, among healthcare workers [6–10]. Key interventions introduced includes: routine screening of staff for blood pressure, blood glucose, and BMI (especially before annual leave); the establishment of staff wellness clinics within hospitals to provide counselling, monitoring, and early treatment of NCDs; lifestyle promotion through health education campaigns on nutrition, exercise, stress management, and avoidance of alcohol and tobacco; workplace support systems integrating occupational health with psychosocial support to address stress and burnout; and structured referral and follow-up care for employees diagnosed with chronic conditions [4–10]. Despite these efforts, gaps remain in the utilization of wellness services, preventive screening, and adherence to lifestyle recommendation by most hospital staff in Ghana [3,10]. Nurses are not exempt from this non-utilization of health services to manage their conditions. Due to long working hours, high job

stress, irregular meal patterns, and sedentary lifestyles, nurses are at increased risk of developing hypertension compared to the general population yet most of them are reluctant to seek care [3,7,8].

Nurses in rural Ghana constitute the largest proportion of healthcare providers [9–12]. They face unique occupational stressors including long shifts, high workloads, and irregular eating habits, which predispose them to hypertension [10,12]. Yet, there remain gaps in understanding how nurses in rural Ghana perceive and respond to their diagnosis [9,11], what barriers they encounter in accessing healthcare, and how social and cultural factors influence their management choices [2,12]. Although the Ghana Health Service introduced the Employee Health and Wellbeing Program to strengthen screening, counselling, and lifestyle modification among healthcare staff [10], evidence suggests limited uptake of these interventions. Reports from the Kintampo Municipal Hospital Wellness Clinic (unpublished internal report, 2021–2022) show a rising trend of hypertension among nurses despite the programs' availability.

## Aim

We explored the health seeking behaviors of nurses diagnosed with hypertension and providing health care in a resource-constrained setting in a rural part of Northern Ghana.

## Theoretical framework underpinning the study

The study was underpinned by the Health Belief Model. The Health Belief Model (HBM) is a psychological framework developed in the early 1950s by Irwin Rosenstock and his colleagues to explain and predict health-related behaviors, particularly in relation to the uptake of health services. This model is particularly pertinent to the study of health-seeking behavior among nurses with hypertension at Kintampo Municipal Hospital in the Bono East Region of Ghana

Key Components of the Health Belief Model includes:

Perceived Susceptibility: This component refers to an individual's belief regarding their risk of developing a health issue. For nurses with hypertension, recognizing their vulnerability to serious complications such as heart disease or stroke can motivate them to seek medical care. Understanding their susceptibility is essential for developing proactive health strategies.

Perceived Severity: This aspect reflects the belief about the seriousness of a health condition and its potential consequences. Nurses who acknowledge the severe outcomes associated with uncontrolled hypertension are likely to be motivated to take appropriate action. Emphasizing these potential risks can encourage healthcare professionals to prioritize their health.

Perceived Benefits: This component focuses on the perceived advantages of adopting health-promoting behaviors. Highlighting the benefits of seeking treatment, such as improved health status and enhanced quality of life, can motivate nurses to engage in health-seeking behaviors. If they believe that treatment significantly reduces their health risks, they are more likely to act accordingly, and vice versa.

Perceived Barriers: Identifying and addressing obstacles that hinder health-seeking behavior is crucial. These barriers may include time constraints, workplace stigma and limited access to healthcare services. Reducing these barriers is essential to increase the likelihood that nurses will seek the necessary medical attention.

Cues to Action: Triggers that prompt individuals to take health-related action. Internal cues, such as symptom experience, and external cues, such as reminders from healthcare providers, can motivate nurses to seek care. Understanding these triggers can assist in developing effective interventions that promote timely health-seeking behavior.

Self-efficacy: This component pertains to an individual's confidence in their ability to successfully perform a behavior-related activity. Enhancing nurses' self-efficacy in managing their hypertension and pursuing care is vital. Providing education and resources can empower them to take proactive steps toward improving their health status.

The Health Belief Model (HBM) is widely utilized in public health to design interventions aimed at promoting health behaviors and increasing the utilization of health services. In the context of this study on nurses with hypertension, applying HBM will yield valuable insights into the beliefs and perceptions that influence their health-seeking behavior. By understanding these factors, targeted strategies can be developed to encourage nurses to prioritize their health and seek necessary medical attention.

The theoretical framework underpinning the study is shown in Fig 1.

## Methods

### Study design

The study employed a qualitative research approach, renowned for its capacity to gather comprehensive information pertinent to the health-seeking behaviors of nurses living with hypertension. This approach facilitated an in-depth understanding of the experiences of nurses and the contextual factors influencing their health [13].

### Study setting

The study was carried out at the Kintampo Municipal Hospital in Kintampo in the Northern part of Ghana in a region known as Bono Region. The choice of Kintampo Municipal Hospital as the study setting was for two key reasons; Firstly, it has an institutionalised wellness clinic where nurses undergo routine screening for blood pressure, glucose monitoring and other vital signs prior to their annual leave. This system provides a unique opportunity to study health-seeking behaviours among nurses living with hypertension using readily available clinical data. Secondly, the facility acts as the regional hospital hence the need to engage its nurses to investigate the impact of hypertension on the job they do and their submission on coping mechanisms and recommendations to use to make informed decisions or policies to help nurses in Ghana and the diaspora at large. The region is ideally situated to investigate health-seeking behaviours in a dynamic and representative environment due to its diversified population and function as a link between northern and southern Ghana. Also, the paucity of research in this area presents a chance to fill up knowledge gaps and contribute perspectives that can guide regional healthcare practices and policy.

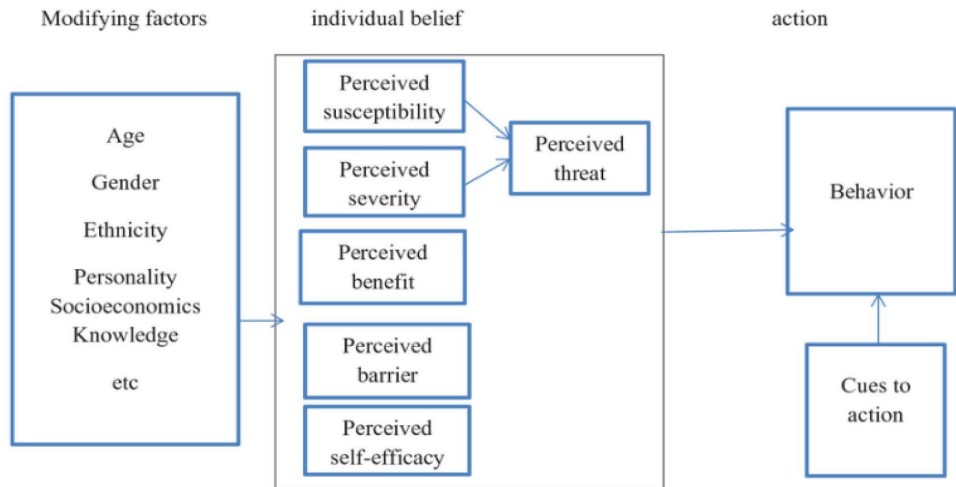

**Fig 1. Health belief model.**

## Target population and sample size determination

The study population was registered nurses who were at post during the data collection and had been diagnosed with hypertension and providing care at the Kintampo Municipal Hospital. This population was purposively selected due to the aim of the study.

The sample size was determined at data saturation which was achieved at the conclusion of the twelfth (12th) interview. We observed data redundancy after the conclusion of the ninth interview and after concluding three additional interviews, we observed that no new information was being elicited hence data saturation at the 12th interview.

## Inclusion and exclusion criteria

To ensure a focused and relevant participant pool, the following inclusion criteria were applied.

- Professional Status: Participants must be nurses currently at Kintampo Municipal Hospital. This criterion ensured that the experiences shared are pertinent to the specific healthcare environment being studied, and that participants have sufficient professional backgrounds to provide insightful perspectives on managing hypertension.

- Health Status: Participants had a confirmed diagnosis of hypertension. This criterion was critical for exploring specific health-seeking behaviours and experiences related to managing chronic conditions.

- Duration of Diagnosis: Participants should have been diagnosed with hypertension for at least six months prior to the study. This duration allowed for a more comprehensive understanding of their ongoing experiences and management strategies.

- Willingness to Participate: Participants must have expressed a willingness to engage in the study and provide informed consent. This criterion ensured that the individuals involved are motivated to share their experiences and insights openly.

- Age: Participants were at least 18 years old. This criterion ensured that all participants are legally able to provide consent and articulate their experiences competently.

We excluded the following:

- Non-Nursing Staff: Individuals who were not nurses or were employed in non-nursing roles at Kintampo Municipal Hospital but with hypertension were excluded.

- Recent Diagnosis: Participants who had been diagnosed with hypertension for less than six months prior to the study were excluded. This criterion was necessary to ensure that participants had adequate time to experience and adapt to their condition, providing richer data.

- Serious Comorbidities: Individuals with serious comorbidities or chronic illnesses that significantly impacted their health status and may confound the study findings were excluded.

- Psychiatric Conditions: Participants with active psychiatric conditions that could impair their ability to communicate their experiences effectively were excluded.

- Temporary or Contract Staff: Nurses employed on a temporary or contract basis for less than three years at the hospital were excluded. This ensured that all participants had a stable employment history and a deeper understanding of the hospital's environment and culture.

- Pregnant nurses with pregnancy induced hypertension: Professional nurse with many years of experience who were experiencing hypertension because they were pregnant were not considered.

**Selection of participants and data collection**

Participants were selected based on their understanding of hypertension through purposive sampling methods [14]. This approach aimed to identify individuals who could offer valuable insights into the perceptions of health-seeking behaviors among nurses with hypertension at Kintampo Municipal Hospital. To implement this method, we arranged meetings with both the Medical Superintendent and the Director of Nursing Services (DNS). This collaboration was essential, as it facilitated access to data concerning nurses with hypertension through the mandatory medical screenings conducted for all nurses prior to their annual leave at the hospital's wellness clinic. Engaging with these officials aided in identifying potential participants who could share their experiences and perceptions regarding health-seeking behaviors, thereby enriching the qualitative data for the study. The data collection took place between 10th October 2025–20th October 2025. We obtained the list of eligible participants and then contacted them to be part of the study after providing them with adequate information about the study as provided in the participant information sheet. We informed the potential participants that participation in the study was voluntary and that the study was solely for academic purposes only. Each participant signed a written informed consent form before taking part in the study.

Data collection took place in one of the consulting rooms at the Outpatient Department (OPD) of the study setting which had been assigned to the research team. Only the research team and the participants had access to the said consulting room during the data collection and management of the hospital were not involved in the data collection even though they gave permission for the collection of data from the eligible participants. We ensured that the management of the study setting did not have access to the consulting room to make the participants comfortable taking part in the study.

Data collection was conducted using pre-tested semi-structured interview guide in English (S1 File), allowing for flexibility in responses while ensuring coverage of key topics. The interview questions were thoughtfully crafted in advance to address specific research issues, fostering detailed insights from participants. Both notetaking and voice recording were utilized to capture participants' exact words, facial expressions and mood.

The interviews lasted between 35–45 minutes and were conducted in English (the official language of communication in Ghana). The interviews were recorded using an audio recorder, transcribed verbatim after each interview. We further did member checking with the participants to ensure their exact views had been captured.

**Data management**

In ensuring the integrity, confidentiality, and accuracy of the research findings, robust data management practices were implemented throughout the study. Participants were given special codes by the research team members, who ensured that they were not coerced but rather allowed to participate voluntarily. Participants were assured of confidentiality, and it was reiterated that the data (recordings, field notes and other recorded observations) were to be used solely for the agreed research purposes and for no other reason. To safeguard the confidentiality of participants, all audio recordings, field diaries with field notes and transcripts were stored securely on a password protected laptop of the 1st author.

**Methodological rigor**

Ensuring methodological rigor was essential for the credibility and reliability of the study findings. This section outlines the strategies that were employed to enhance the rigor of the study:

**Credibility.** To establish the credibility of the findings, the following strategies were employed:

**Member Checking:** Participants were invited to review the transcripts of their interviews and the initial findings. This process allowed them to confirm the accuracy of the data and provided an opportunity to clarify or expand upon their responses, which is a recognized method for enhancing credibility in qualitative research.

**Prolonged Engagement:** We spent sufficient time with participants during the interview process, fostering rapport and encouraging open dialogue. This engagement helped to deepen the understanding of the participants' experiences, as prolonged engagement is crucial for building trust and obtaining rich data.

**Peer Debriefing:** Regular discussions with colleagues who had expertise in qualitative research were conducted. This external perspective helped identify potential biases and enhanced the interpretation of the data, aligning with best practices in qualitative research.

**Transferability enhancements.** To enhance transferability, the study provided rich, detailed descriptions of the research context, participants, and findings. This included:

**Thick Description:** Comprehensive narratives about participants' backgrounds, the setting of the study, and the nuances of their experiences were documented. This allows readers to assess the relevance of the findings to other contexts, which is essential for qualitative research.

**Contextual Factors:** We discussed how cultural, social, and institutional factors influenced the experiences of nurses living with hypertension, providing a framework for understanding the findings in different settings.

**Dependability.** To ensure dependability, the following measures were implemented:

**Audit Trail:** A detailed log of the research process, including decisions made during data collection and analysis, was maintained. This transparency allowed for external scrutiny and enhanced the reliability of the study, as recommended in qualitative research methodologies.

**Consistent Procedures:** Standardized protocols for conducting interviews, data analysis, and managing data were followed to minimize variability and maintain consistency throughout the research process.

**Conformability.** To enhance conformability and reduce researchers' bias, the following strategies were employed:

**Reflexivity:** We engaged in reflexivity by maintaining a reflective journal throughout the research process. This journal documented our personal thoughts, feelings, and assumptions that may have influenced the research, helping to identify and mitigate biases, a critical aspect of qualitative research.

**Triangulation:** Although this study relied solely on in-depth interviews, triangulation was achieved through multiple strategies: data source triangulation (interviewing nurses of different ranks and departments), and investigators' triangulation (engaging more than one researcher in coding and analysis), These approaches enhanced the trustworthiness and credibility of the study findings.

## Ethical approval and consent to participate

We complied strictly with the declaration of Helsinki regarding data collection. The study was approved by the Kintampo Research Center's Scientific Review Committee (Protocol Number: khrc/adm/14/2025-10) and institutional approval was sought from the study setting before participant recruitment and data collection. We obtained permission from the Unit Head of the Out-patient Department of the Staff Wellness Clinic of the study setting prior to data collection. We informed the participants that participation in the study was voluntary and that refusing to participate had no impact on their status as employees. Also, we explained the purpose of the study to each participant before obtaining written informed consent for the study. Further, we ensured confidentiality of the data collected by assigning unique codes to each response. We also maintained privacy and ensured anonymity throughout the data collection and analysis of the data. During the data collection, participants who appeared psychologically distressed were referred to a clinical psychologist in the facility to help make them stable.

## Data analysis

Thematic analysis was used to analyze the qualitative data with the aid of Nvivo 14.0. The audio files of the interviews were transcribed verbatim after data collection and the word transcripts as well as the records in the field notebooks/diaries were used in the data analysis. We embarked on concurrent thematic analysis with data collection in which we interview was analyzed before proceeding to the next interview. The process of the thematic analysis involved familiarization with the transcripts, systematic coding, and the development of themes. The NVivo application was chosen because it is particularly suitable for qualitative data analysis. Importantly, the Health Belief Model (HBM) served as the analytical framework. Codes and themes were mapped against the HBM constructs of perceived susceptibility, perceived severity, perceived benefits, perceived barriers, cues to action, and self-efficacy. This approach ensured that the analysis was both data-driven and theory-informed, thereby allowing the study to generate findings that were grounded in participants lived experiences while remaining aligned with a robust theoretical model.

## Results

A total of twelve (12) participants were recruited for this study; nine (9) females and three (3) males. The participants' ages ranged from twenty-nine to fifty-nine (29–59) years, with a mean age of approximately 39 years. The cohort represented a range of professional seniority: three (3) were Principal Nursing Officers, four (4) were Senior Nursing Officers, two (2) were Staff Nurses (including one Staff Midwife), and one (1) each held the roles of Midwifery Officer, Senior Staff Nurse, and Rotation Nurse. Regarding professional experience, the participants' years in service varied widely. One (1) participant was a newly recruited Nurse with eight months of working experience in the facility, three (3) participants had one to five (1–5) years of experience, four (4) had six to ten (6–10) years, three (3) had eleven to eighteen (11–18) years, and one (1) highly experienced participant had forty-two (42) years of service.

The themes and sub-themes that were generated from the data is shown in Table 1.

### Perception of the severity of hypertension

Understanding nurses' perceptions of the severity of hypertension provides insight into how they interpret their personal health risks and the seriousness they attach to this chronic condition. Perceived severity, a core construct in health behavior theories such as the Health Belief Model, reflects an individual's belief about the extent of harm that may result from a disease and its potential consequences on daily life and well-being. In this study, exploring nurses' perceptions helps to reveal how their awareness, professional knowledge, and personal experiences shape their attitudes toward prevention, lifestyle modification, and treatment adherence regarding hypertension.

**Biomedical understanding and awareness of hypertension.** This subtheme captured nurses' biomedical understanding and awareness of hypertension as a chronic, non-communicable condition. It explores their knowledge of the disease's etiology, risk factors, and physiological implications, as well as their recognition of its silent and progressive nature. Given their clinical background, nurses are expected to possess a higher level of biomedical literacy about hypertension. However, variations in understanding often influence how they perceive personal vulnerability and the seriousness of the condition. This subtheme, therefore, highlights how nurses conceptualize hypertension from a professional and health-oriented perspective, and how this knowledge informs their attitudes toward prevention and self-care practices.

Most nurses demonstrated a clear biomedical understanding of hypertension as a chronic and clinically significant condition. They commonly describe it as a persistent elevation of blood pressure that exerts stress on the heart and increases the risk of complications such as stroke. As one nurse stated:

*"Hypertension is when the normal BP goes beyond the normal range… when the heart cannot pump blood effectively." (Nurse 1)*

**Table 1. Organization of major themes and sub-themes.**

| Main Theme | Sub-theme |
| --- | --- |
| **Understanding and Perception of Hypertension** | Biomedical Understanding of the Condition |
| | Perceived Severity Among Nurses |
| | Occupational Risk Perception (Work as a Causal Factor) |
| **Knowledge Acquisition and Information Sources** | Professional and Formal Sources (Workshops, Colleagues, Journals) |
| | Digital and Informal Sources (Internet, Google, Social Media) |
| | Lack of Formal Institutional Guidance |
| **Impact on Work Performance and Well-being** | Physical Effects (Fatigue, Weakness, Headaches, Palpitations) |
| | Cognitive and Emotional Impact (Reduced Focus, Anxiety) |
| | Perceived Effect on Patient Care and Safety |
| **Self-Management and Coping Strategies** | Adherence to Antihypertensive Medication |
| | Lifestyle Modifications (Healthy Diet, Exercise, Rest) |
| | Self-Monitoring Practices (Routine BP Checks) |
| | Use of Complementary or Faith-Based Coping Strategies |
| **Barriers to Healthcare Access and Management** | Work-Related Structural Barriers (Shift Work, Heavy Workload, Time Constraints) |
| | Institutional Neglect (Absence of Staff Wellness Policies) |
| | Financial Constraints (Medication and Clinic Costs) |
| | Psychosocial Barriers (Confidentiality Concerns, Self-Reliance) |
| **Organizational Culture and Institutional Support** | Dominant Culture of Patient-Care Over Staff Well-being |
| | Limited Supervisory or Administrative Support |
| | Supportive Peer Interventions (Task Adjustment, Reminders) |
| **Stigma and Professional Identity** | Fear of Being Perceived as Weak or "Unfit" for Duty |
| | Experience of Stigma, Gossip, or Judgment in the Workplace |
| | Self-Reliance and Professional Resilience Norms |
| **Social Support Systems** | Family Support (Spousal, Parental, Sibling Encouragement) |
| | Peer Support and Team Understanding |
| | Limited or Absent Social Support Networks |
| **Recommendations for Improvement** | Institutional Wellness Programs (Screenings, Counseling, Health Education) |
| | Policy and Structural Reforms (Reduced Shifts, Subsidized Medications) |
| | Establishment of Peer Support Groups and Safe Discussion Spaces |

Another emphasized its long-term danger, stating:

*"Hypertension is when your BP is constantly high, and if not treated, it can lead to stroke."* ***(Nurse 3)***

**Perceived severity among nurses.** This subtheme explores how nurses perceive the seriousness of hypertension, both as a clinical condition and a personal health threat. It examines their awareness of its long-term consequences, such as stroke, heart failure, and renal disease, and how these perceptions influence their sense of urgency toward monitoring and managing their own blood pressure. Despite their medical knowledge, perceptions of severity often vary based on personal experiences, exposure to patients with complications, and perceived control over the condition.

Nurses perceived hypertension as a serious and growing health concern, especially among female healthcare workers. They attributed its prevalence to the demanding and stressful nature of their profession.

As one nurse observed:

*"It's very severe, very, very severe, especially among the female nurses."* ***(Nurse 9)***

Another added:

*"Because of our job, we easily get stressed out, and it's very severe among us." (Nurse 6)*

**Work-related stress as a risk factor.** This subtheme examined nurses' perceptions of work-related stress as a significant contributor to the development and worsening of hypertension. Given the demanding nature of nursing, characterized by long shifts, high patient loads, emotional strain, and limited recovery time, many participants identified occupational stress as both an unavoidable part of their profession and a critical health risk. The narratives highlight how sustained exposure to stressful work environments influences blood pressure levels, coping behaviors, and overall cardiovascular health. This subtheme also reflects on how institutional pressures, inadequate staffing, and emotional fatigue intersect to shape nurses' recognition of stress as a modifiable yet pervasive risk factor for hypertension. Workload, night shifts, and emotional strain were cited as major stressors influencing hypertension severity.

According to the participants, the primary sources of stress among nurses stem from their demanding work environment, including night shifts and the emotional strain of managing patients' conditions. One nurse explained:

*"The hospital itself stresses us… standing there, walking about." (Nurse 11)*

Another added:

*"The stress, night shifts, and patient conditions can trigger it." (Nurse 10)*

**Perceived impact on work performance.** This subtheme explored how nurses perceive the effects of hypertension on their professional performance and capacity to deliver quality patient care. Participants described how symptoms such as fatigue, headaches, dizziness, and emotional instability occasionally hinder their concentration, decision-making, and physical endurance during clinical duties. For some, fear of health deterioration led to heightened anxiety or self-imposed limitations in workload, while others reported reduced efficiency and productivity during hypertensive episodes. The accounts reveal dual awareness: nurses recognize the importance of maintaining optimal health to ensure patient safety, yet they also grapple with balancing personal health needs against the demanding expectations of their work environment. Perceptions varied; while some nurses minimized the impact of hypertension on their work, others admitted it affected concentration and stamina.

The participants expressed mixed views on how hypertension affects their work performance. One nurse noted that the condition's symptoms, such as severe headaches, could interfere with focus and caregiving, stating:

*"When you have severe headaches, you tend to pay attention to yourself more than others." (Nurse 11)*

However, another participant disagreed, suggesting minimal interference with job performance:

*"I don't see it affecting my work in any way." (Nurse 4)*

### The meanings nurses attribute to the risks associated with hypertension

This theme focused on how nurses conceptualize and internalize the risks associated with hypertension, both as healthcare professionals and as individuals vulnerable to the condition. Participants' narratives revealed a complex blend of biomedical knowledge, personal experience, and professional observation shaping their understanding of hypertension's consequences. Many nurses viewed hypertension not merely as a chronic medical condition but as a silent and potentially life-threatening disease with

serious implications such as stroke, heart failure, and kidney dysfunction. Others emphasized the emotional and social dimensions of risk, particularly the fear of sudden incapacitation, premature death, or inability to fulfil professional and family responsibilities.

## Understanding of risks and causation

This theme focused on nurses' understanding of the factors that contribute to the onset and progression of hypertension, highlighting how they interpret the interplay between biological, behavioral, and occupational risks. Participants demonstrated varying levels of biomedical knowledge regarding hypertension's etiology, many accurately linked it to lifestyle factors such as poor diet, physical inactivity, obesity, and genetic predisposition. However, their accounts also reflected contextual and experiential interpretations, with several nurses attributing causation to persistent work-related stress, irregular meal patterns, and demanding shift schedules typical of the nursing profession. This understanding of causation appeared to shape their personal sense of vulnerability and informed their preventive health behaviors. Overall, the findings suggest that nurses' conceptualizations of hypertension risk are influenced not only by clinical knowledge but also by their lived experiences within the healthcare system.

**Biomedical and occupational risks.** Nurses described hypertension as resulting from both biomedical and occupational factors. They linked the condition to genetics, poor diet, aging, and lack of exercise, showing sound biomedical understanding. At the same time, they identified long working hours, heavy workloads, emotional stress, and irregular shifts as key occupational risks that increase their vulnerability. This shows that nurses recognize hypertension as both a medical and work-related condition shaped by their demanding professional environment. Participants linked hypertension to cardiovascular risk, stroke, and occupational stress inherent in nursing. Participants demonstrated awareness of both biomedical and occupational risks associated with hypertension. They recognized that uncontrolled high blood pressure could result in severe health outcomes such as stroke and cardiovascular disease, as expressed by one nurse:

*"When your BP is high and not treated, it can lead to stroke and cardiovascular disease." (Nurse 3)*

Additionally, nurses acknowledged the demanding nature of their work as a contributing factor, with one stating,

*"The work is stressful; sometimes you even forget to eat or rest." (Nurse 2)*

**Lifestyle and personal risk factors.** Nurses acknowledged that their lifestyle choices contribute to the risk of developing hypertension. They identified poor dietary habits, inadequate physical activity, lack of rest, and excessive intake of stimulants such as caffeine as common risk factors. Many admitted that busy work schedules and fatigue often lead to neglect of healthy routines. This reflects an awareness that personal behaviors, alongside occupational pressures, play a significant role in hypertension risk. Diet, irregular mealtimes, lack of exercise, and sleep deprivation were perceived as risk amplifiers. Nurses acknowledged the role of lifestyle and personal habits in contributing to hypertension risk. They emphasized the importance of maintaining healthy eating practices, as one noted:

*"You shouldn't be eating too much of fatty foods." (Nurse 12)*

However, their demanding work schedules often led to unhealthy eating patterns, such as late meals after shifts. As one participant explained:

*"When you come for afternoon shift, you close late, and that's when you eat." (Nurse 6)*

**Knowledge sources.** Nurses reported acquiring knowledge about hypertension from multiple sources, including formal training, workplace health education sessions, and professional experience in patient care. Some also mentioned learning through personal experiences and media platforms. However, the depth of understanding varied, with a few relying mainly

on experiential knowledge rather than evidence-based information. This indicates that while awareness exists, continuous professional education is essential to strengthen accurate and up-to-date knowledge about hypertension. Nurses drew information from formal and informal sources including textbooks, the internet, and peer learning. The participants identified various sources of knowledge about hypertension, ranging from academic materials to digital platforms. As one nurse mentioned:

*"I read from books like Rosen and Rosen… sometimes from social media like ChatGPT and Google." **(Nurse 11)***

Another participant said:

*"Mostly the internet and online libraries." **(Nurse 10)***

**Perception of personal vulnerability.** Most nurses acknowledged that their profession exposes them to high stress and irregular routines, which increase their risk of developing hypertension. However, perceptions of personal vulnerability varied; while some recognized themselves as at risk due to family history or workload, others felt protected by their medical knowledge or healthy lifestyles. This mixed perception reflects a gap between awareness of occupational risk and personal risk acknowledgement.

Some nurses recognized their personal vulnerability to hypertension, linking it to emotional distress, heredity, or physiological changes such as pregnancy. One participant shared:

*"Not quite long I lost my husband… I think that's the reason why mine is still there."*

***(Nurse 5)***

Another explained:

*"Some is due to stress, some to family history, some to lifestyle." **(Nurse 8)***

### The cultural factors influencing nurses' health management choices

Cultural beliefs and social norms played a significant role in shaping how nurses manage hypertension and other health conditions. The participants reported relying on both biomedical treatment and traditional remedies, reflecting a cultural inclination toward pluralistic health-seeking behavior. Family expectations, religious faith, and community attitudes toward illness also influenced whether nurses sought formal care, practiced self-medication, or delayed treatment. These cultural influences highlight the complex interplay between professional knowledge and deeply rooted social values in nurses' health management decisions.

### Cultural and personal influences on hypertension management

This theme focused on how cultural beliefs, personal values, and social expectations shape nurses' approaches to managing hypertension. Despite their medical knowledge, the nurses reported being influenced by cultural norms surrounding illness, traditional healing practices, and perceptions of strength and resilience. Personal beliefs about faith, family responsibilities, and community expectations often guided decisions on treatment, lifestyle modification, and adherence to medication. The findings reveal that hypertension management among nurses is not solely biomedical but intertwined with cultural identity and personal meaning.

**Self-reliance and privacy norms.** The nurses emphasized a strong sense of self-reliance and the need to maintain privacy regarding their health conditions. They often preferred to manage hypertension independently rather than disclose it to colleagues or seek institutional support. This tendency stemmed from professional pride, fear of stigma, and cultural

expectations that health workers should appear strong and capable. As a result, some nurses delayed seeking medical attention or downplayed symptoms, reflecting how privacy norms can influence health-seeking behavior and self-management practices.

Many participants highlighted a preference for self-management and maintaining confidentiality about their health conditions. As one nurse stated:

*"I'm someone who doesn't normally talk to people about my stuff." **(Nurse 1)***

Another added:

*"Maybe because I don't really discuss with them; I manage my own thing." **(Nurse 5)***

**Professional identity and endurance culture.** Nurses described a strong professional identity built around resilience, dedication, and the expectation to "endure" challenges without complaint. This endurance culture often discouraged them from prioritizing their own health, as attending to personal illness was seen as a sign of weakness or lack of commitment to patient care. Consequently, some nurses continued working despite hypertension-related symptoms, reinforcing a workplace norm where self-care was secondary to professional duty.

Nurses expressed a deep sense of professional responsibility, frequently placing patients' needs above their own well-being. One participant remarked:

*"You can't use hypertension as an excuse; if all of us start, nobody will come for night shifts." **(Nurse 7)***

Another added:

*"Sometimes I come to work not well; we tend to sacrifice our own health for patients." **(Nurse 4)***

**Belief in personal responsibility and faith-based approaches.** Many nurses viewed managing hypertension as a personal responsibility, emphasizing discipline in lifestyle choices and adherence to medication. Alongside this, faith played a significant role in their coping strategies, several attributed their health outcomes to divine will or relied on prayer for strength and healing. This combination of self-reliance and spiritual belief shaped their approach to managing hypertension, often complementing or at times, substituting formal medical care.

Some nurses viewed hypertension management as a matter of personal discipline, compliance, and spiritual strength. One participant explained:

*"You should take your medications and listen to what doctors say." **(Nurse 2)***

Another added:

*"They should organize people in the community and educate them about hypertension." **(Nurse 1)***

This highlights the importance of awareness and collective responsibility. These perspectives reflect a blend of personal accountability and faith-based endurance in managing chronic illness.

## The impact of social support on nurses' health-seeking actions

Social support emerged as a critical factor influencing nurses' willingness and ability to seek care for hypertension. Encouragement from family, colleagues, and close friends often motivated timely medical consultations and adherence to

treatment. Conversely, limited support or fear of judgment from peers led some nurses to delay care or manage symptoms privately. The presence of understanding supervisors and supportive workplace networks enhanced openness to seeking help, underscoring the importance of social connectedness in shaping nurses' health-seeking behaviors.

**Role of social and organizational support**

This theme highlights how interpersonal and institutional support systems shape nurses' approaches to managing hypertension. Support from family, colleagues, and supervisors fosters positive health-seeking behaviors and adherence to care, while the absence of such support contributes to neglect or self-management. Organizational backing, such as accessible health services, flexible schedules, and an encouraging work culture, plays a pivotal role in enabling nurses to prioritize their own health alongside professional responsibilities.

**Workplace and peer support.** This subtheme focused on how encouragement and understanding from colleagues and supervisors influence nurses' health-seeking behaviour. Supportive work environments, where peers share experiences, remind one another about checkups, or provide emotional backing- help normalize seeking care for hypertension. Conversely, unsupportive settings or judgmental attitudes can discourage disclosure and delay treatment.

Some nurses described supportive peer relationships that fostered adherence and reduced work-related strain. One of participants noted:

*"They advise me to take my medications as I'm always supposed to."* ***(Nurse 1)***

Another shared:

*"My colleagues sometimes exempt me from certain things."* ***(Nurse 3)***

**Family support and emotional encouragement.** This subtheme highlights the crucial role families play in motivating nurses to manage their hypertension. Emotional encouragement, practical assistance, and concern from spouses or relatives often prompt nurses to seek care and adhere to treatment. In contrast, limited family support or misunderstanding of the condition can lead to neglect of health needs and reduced motivation for ongoing management.

Family members were described as crucial sources of motivation and practical support in managing hypertension. One nurse stated:

*"My family helps by reminding me to take my drugs and helping me choose the right diet."* ***(Nurse 3)***

Another added:

*"They help me with my exercise and diet plan."* ***(Nurse 6)***

**Institutional support and gaps.** This subtheme explored how institutional structures influence nurses' ability to manage hypertension effectively. While some nurses acknowledge supportive policies such as periodic screenings and health education programs, many identify gaps such as inadequate staff wellness initiatives, limited access to medical checks, and lack of flexible work schedules. These institutional shortcomings often discourage proactive health-seeking and reinforce a culture of neglecting personal health among nurses. Across all interviews, participants highlighted the absence of structured institutional wellness systems that cater to nurses' health needs. One participant explained:

*"As for my institution, it's just the medication you have to take as prescribed."* ***(Nurse 2)***

While another added:

> *"Nothing at all… we are even expected to do more." **(Nurse 1)***

**Perceived stigma and judgment.** This subtheme highlights nurses' concerns about being judged or stigmatized if they disclose their health conditions, particularly hypertension. Many nurses fear that admitting illness may be perceived as a sign of weakness or incompetence within the profession. This perception often discourages them from seeking care openly or discussing their condition with colleagues and supervisors, ultimately contributing to delayed treatment and poor health management practices. A recurring barrier was stigma from patients and colleagues who viewed nurses as "immune" to illness. Some nurses expressed awareness of the stigma surrounding healthcare workers who experience illness, noting that patients and the public often hold unrealistic expectations of them. Participants shared that when nurses themselves develop conditions like hypertension, others perceive it as contradictory to their professional role. As one nurse remarked:

> *"Sometimes patients say, 'Hey, you too are taking some?' They think you shouldn't have such a thing." **(Nurse 2)***

Another added:

> *"People see health professionals as gods… how can you treat disease and you yourself have it?" **(Nurse 7)***

### The barriers encountered when accessing healthcare services

This section explored the various obstacles nurses face when attempting to seek medical care for hypertension. Participants described barriers such as heavy workloads, long and inflexible shifts, and limited time to attend clinic appointments. Others cited financial constraints, long waiting times, and perceived discrimination when seeking care in their own facilities. Additionally, some nurses expressed reluctance to seek care due to confidentiality concerns, fear of being recognized by patients or colleagues, and a tendency to prioritize patient needs over their own health. These barriers collectively hinder timely diagnosis, treatment adherence, and consistent follow-up for hypertension management.

### Barriers to healthcare access and management

This theme captures the multifaceted challenges that prevent nurses from effectively accessing and managing healthcare for hypertension. Despite their medical knowledge, participants described structural, organizational, and personal barriers that limit their ability to seek timely and consistent care. Heavy workloads, staffing shortages, and long shifts often leave little time for self-care or clinic visits. Financial limitations, bureaucratic delays, and inadequate workplace health policies further exacerbate the problem. Additionally, psychological and social barriers, such as fear of stigma, concerns about confidentiality, and the belief that healthcare workers should appear strong, discourage many from seeking help.

**Workload and scheduling constraints.** Participants identified their demanding work schedules as a major barrier to seeking healthcare. Long shifts, unpredictable duty rosters, and staff shortages made it difficult to find time for medical appointments or regular monitoring. Many nurses admitted postponing check-ups or medication refills due to exhaustion or competing work demands. This constant pressure to prioritize patient care over personal health often led to neglect of their own hypertension management.

Nurses described how demanding work schedules and inadequate rest periods made it difficult to maintain consistent healthcare routines. The pressure of shift work and limited flexibility often prevented them from seeking timely care or adhering to prescribed treatments. As one nurse explained:

*"You're not given time to sleep during work… not given the days off you need." (Nurse 3)*

Another participant mentioned:

*"We don't get time to take our medications on time as expected." (Nurse 7)*

**Financial and systemic barriers.** Some nurses reported that the cost of healthcare services, medications, and follow-up visits posed a challenge, especially when not fully covered by insurance. Delays in salary payments and inconsistent hospital support for staff welfare worsened the situation. Additionally, systemic issues such as long waiting times, bureaucracy, and limited availability of essential drugs discouraged consistent health-seeking and treatment adherence among nurses.

Nurses emphasized that financial limitations and inadequate insurance coverage significantly affected their ability to manage hypertension effectively. The high cost of essential medications often forced them to compromise on treatment quality or delay care. One nurse stated that*:*

*"Some medications are not covered by insurance, so we end up using low-quality drugs." (Nurse 3)*

Another had this to say:

*"Sometimes financial constraints can also be a contributing factor." (Nurse 4)*

**Psychosocial and cultural barriers.** Psychological factors such as fear of diagnosis, denial, and perceived stigma often discouraged nurses from seeking timely care. Some preferred self-management or traditional remedies to avoid being viewed as weak or unfit for duty. Cultural beliefs emphasizing endurance and self-reliance further limited open discussion and proactive management of hypertension among nurses.

Concerns about self-image and workplace gossip discouraged nurses from openly seeking care for hypertension. The fear of being judged or having their condition disclosed to colleagues made some prefer silence over support. A nurse shared:

*"Some will go and tell them I am not coming to work when my BP is high." (Nurse 5)*

**Lack of institutional health infrastructure.** Many nurses highlighted the absence of accessible, nurse-centered health services within their institutions. Limited screening programs, inadequate staff clinics, and lack of confidential health support systems made it difficult to manage hypertension effectively. This infrastructural gap reinforced delayed care-seeking and self-treatment practices among nurses.

Nurses highlighted the absence of institutional structures that promote their health, such as wellness units, dietary support, and regular medical screening. These gaps limit opportunities for preventive care and recovery within the workplace. As expressed by a nurse:

*"It should be like a system where they provide a small canteen or a kitchen so we can eat and rest." (Nurse 9)*

## Summary of findings

Overall, nurses at Kintampo Municipal Hospital perceive hypertension as a severe yet manageable chronic condition. Their health-seeking behavior is shaped by self-reliance, occupational stress, and weak institutional support structures. Despite high awareness and knowledge, barriers such as stigma, workload, and inadequate rest time hinder regular healthcare utilization. Participants emphasized the urgent need for workplace wellness initiatives, flexible schedules, and psychological support systems for hypertensive staff.

## Discussion

The study revealed that nurses possessed a clear biomedical understanding of hypertension as a chronic condition characterized by persistently elevated blood pressure. Participants recognized hypertension as a "silent killer," often asymptomatic until complications occur, reflecting high health literacy consistent with professional knowledge. This aligns with findings from similar studies in Ghana and Nigeria, which noted that nurses are typically well-informed about hypertension due to their medical training and exposure to clinical practice [15,16]. Despite this knowledge, hypertension was widely perceived as highly prevalent and severe among nurses, particularly females. Participants attributed this to occupational stress, long working hours, irregular meal patterns, and emotional fatigue, echoing prior evidence that links nursing work environments to elevated stress-induced blood pressure [17,18]. The finding supports the perceived severity construct of the Health Belief Model, suggesting that nurses recognize the grave consequences of hypertension but still experience challenges translating awareness into preventive action. However, the perception of severity did not always translate into perceived vulnerability. Some participants downplayed the impact of hypertension on work performance, asserting it "does not affect their work," while others admitted to fatigue, headaches, or reduced concentration. This divergence suggests cognitive dissonance between professional knowledge and lived experience; a phenomenon observed in studies among healthcare workers who normalize illness as part of the job [19]. Such normalization may impede timely health-seeking among the participants and could lead to aggravated consequences and complications [1,10,15].

The participants associated hypertension with serious health risks including stroke, heart failure, and premature death. They also recognized occupational and lifestyle risk factors, such as stress, irregular meals, and sleep deprivation, mirroring previous research showing that healthcare professionals, despite awareness, often engage in unhealthy behaviors due to workplace demands [10,20]. The nurses' sources of knowledge, professional workshops, WHO and Ghana Health Service guidelines, online databases, and peer consultation, demonstrated multiple channels of information acquisition. This triangulation of knowledge suggests high cognitive awareness but limited behavioral translation, consistent with earlier findings that knowledge alone is insufficient to alter risk behaviors without institutional support [21,22]. Importantly, some participants cited personal and emotional experiences such as bereavement and family history as major risk amplifiers. Such subjective risk appraisal emphasizes the psychological dimension of health-seeking behavior, where personal experiences often outweigh biomedical instruction. Within HBM, this reflects perceived susceptibility, the belief that one's personal risk is high, which theoretically should motivate action. However, in this study, while susceptibility was acknowledged, structural and occupational barriers inhibited the desired response, demonstrating a classic gap between perception and practice [23,24].

A striking finding was the pervasive culture of self-reliance and silence regarding personal health among nurses. Most participants preferred self-management and rarely disclosed their condition to colleagues, citing privacy and fear of stigma. This resonates with literature suggesting that healthcare professionals often internalize illness, perceiving disclosure as weakness or professional incompetence [19,25]. This culture aligns with what sociologists describe as the "professional invincibility narrative", a social construction that compels nurses to appear resilient, even when unwell [26]. The normalization of illness within the nursing profession may lead to delayed treatment-seeking and poor chronic disease control. Additionally, participants described a strong ethic of endurance and self-sacrifice, prioritizing patient care over self-care. Nurses reported working through illness, skipping rest or medication, and tolerating fatigue to maintain service continuity. Such patterns parallel findings from South Africa and Kenya, where nurses view caring for others as a moral obligation overriding personal well-being [19,27]. From an HBM lens, this can be interpreted as low perceived benefits and high perceived barriers, nurses understand the benefit of seeking care but weigh it against professional expectations and workload pressures. Consequently, their decision to delay care may represent a rational adaptation to a high-demand work environment rather than ignorance or negligence.

The study found mixed experiences regarding social support. While peer and family support were present in some cases, such as reminders to take medication or emotional encouragement, institutional support was largely absent. Participants reported a lack of structured wellness programs, health screenings, and counseling services for hypertensive staff,

consistent with previous Ghanaian studies highlighting poor occupational health frameworks in healthcare institutions [28–30]. Family involvement emerged as a protective factor, consistent with social capital theory which posits that family support enhances adherence and recovery among chronically ill individuals [31]. Family reminders about diet, medication, and rest played compensatory roles in mitigating the absence of institutional health systems. However, such reliance on family highlights systemic neglect of workplace health promotion. Furthermore, nurses described stigma from colleagues and patients, who often viewed healthcare workers as "immune" to illness. This social stigma discouraged open dialogue and disclosure, reinforcing isolation and self-management. Similar patterns have been observed among healthcare workers with mental illness or chronic conditions in sub-Saharan Africa [10,32]. Within the HBM framework, stigma constitutes a perceived barrier that diminishes motivation to seek institutional support, as individuals fear social repercussions more than health consequences.

Workload, inflexible shifts, and fatigue were the dominant barriers to consistent hypertension management. Participants described an inability to rest, irregular meal patterns, and a lack of time for medical appointments, echoing previous studies that identified excessive workload as a major determinant of poor self-care among nurses [6,29–32]. This confirms the HBM's assertion that even when perceived threat and benefit are high, structural constraints can inhibit behavior change. Financial constraints were also noted, particularly regarding the cost of medications not covered by the National Health Insurance Scheme (NHIS). Some nurses resorted to cheaper, less effective alternatives or delayed purchasing prescriptions, corroborating earlier reports of financial strain among Ghanaian healthcare workers managing chronic diseases [33,34]. Psychosocial barriers, including fear of gossip and judgment, also limited help-seeking. Participants described the absence of institutional health infrastructure such as staff clinics, healthy canteens, and wellness days, indicating a policy gap in occupational health. These findings reinforce calls by global health agencies for workplace wellness policies that prioritize preventive and chronic care among healthcare workers [6,30–32].

### Limitations of the study

The study was hospital-based and did not involve other nurses who were not at post or not diagnosed with hypertension. The findings cannot represent the views of all nurses in the hospital. Similar studies which focus on the perspectives of all nurses and other care professionals are encouraged to understand the holistic hypertension situation in the study setting.

The data could have been affected by recall bias even though efforts were made to ensure the participants gave a vivid account during the study.

Furthermore, the findings like most response-based studies could be affected by social desirability issues. However, the researchers asked the participants to give a vivid account of their experiences as per the interview guide and this reduced the social desirability issues.

### Conclusion and recommendations

A significant gap exists between nurses' knowledge of hypertension and their health-seeking actions. This gap is primarily driven by organizational and systemic barriers within the workplace, rather than a lack of individual awareness. To protect this critical workforce, we recommend that nurse managers, hospital management and health policymakers must prioritize the implementation of structured, low-cost workplace wellness programmes. These should include routine screening, flexible scheduling, anti-stigma campaigns to promote health-seeking, and confidential peer support systems to enable nurses to translate their knowledge into consistent self-care practices when diagnosed with hypertension.

### Supporting information

**S1 File. Interview guide.**
(DOCX)

## Author contributions

**Conceptualization:** Kennedy Dodam Konlan, Thomas Ankomanin, Kwadwo Ameyaw Korsah.

**Formal analysis:** Kennedy Dodam Konlan.

**Investigation:** Thomas Ankomanin, Kwadwo Ameyaw Korsah.

**Methodology:** Kennedy Dodam Konlan, Thomas Ankomanin, Kwadwo Ameyaw Korsah.

**Project administration:** Thomas Ankomanin.

**Supervision:** Kwadwo Ameyaw Korsah.

**Validation:** Kwadwo Ameyaw Korsah.

**Writing – original draft:** Kennedy Dodam Konlan.

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
