## [Decision Letter · Decision Letter 0]

26 Dec 2025

Dear Dr. Konlan,

Thank you for submitting your manuscript to PLOS ONE. After careful consideration, we feel that it has merit but does not fully meet PLOS ONE’s publication criteria as it currently stands. Therefore, we invite you to submit a revised version of the manuscript that addresses the points raised during the review process.

We look forward to receiving your revised manuscript.

Kind regards,

Javier Fagundo-Rivera, PhD

Academic Editor

PLOS One

Journal Requirements:

2. We note that your Data Availability Statement is currently as follows: All relevant data are within the manuscript and in Supporting Information files.

**Additional Editor Comments:**

Dear Authors,

Thank you for your work and for your interest in PLOS ONE as a venue to disseminate your research.

Three reviewers have evaluated your manuscript. A major revision is required, addressing the comments from all three reviewers.

Please revise your manuscript accordingly and submit a detailed response-to-reviewers letter indicating how you have addressed each of the reviewers’ comments.

Kind regards,

Reviewers' comments:

Reviewer's Responses to Questions

**Comments to the Author**

1. Is the manuscript technically sound, and do the data support the conclusions?

Reviewer #1: Yes

Reviewer #2: Partly

Reviewer #3: Yes

2. Has the statistical analysis been performed appropriately and rigorously?

Reviewer #1: Yes

Reviewer #2: N/A

Reviewer #3: N/A

3. Have the authors made all data underlying the findings in their manuscript fully available?

Reviewer #1: No

Reviewer #2: Yes

Reviewer #3: Yes

4. Is the manuscript presented in an intelligible fashion and written in standard English?

Reviewer #1: No

Reviewer #2: Yes

Reviewer #3: Yes

**Reviewer #1:  SEE DOCUMENT ATTACHED**

This study, titled "Health seeking behaviors of nurses diagnosed with hypertension and providing health care in resource-constrained setting in a rural part of Northern Ghana: A qualitative study," explores the personal health management strategies and associated challenges faced by nurses living with hypertension in a high-stress, resource-constrained environment in Ghana. The manuscript provides valuable insights into the persistent gap between high professional knowledge and poor health-seeking behavior, attributing this divergence primarily to organizational and systemic barriers. I reviewed the study following PLOS ONE criteria which I now submit.

**Reviewer #2:**

I recommend major revision: (1) sample/ID inconsistency (n=12 but quotes labeled “Nurse 13”); (2) clarify recruitment, voluntariness, and privacy (management involvement; interview location); (3) align the Data Availability statement with PLOS requirements; (4) correct the Limitations section.

**Reviewer #3: SEE DOCUMENT ATTACHED**

The manuscript is scientifically sound and backed by data. The authors need to correct grammar mistakes and try not to mix discussion with results, otherwise the manuscript reflects good scientific rigor and the conclusions and recommendations tie up well with the findings of the study. With minor corrections, it is a good manuscript for publication.

**Do you want your identity to be public for this peer review?** For information about this choice, including consent withdrawal, please see our Privacy Policy

Reviewer #1: No

Reviewer #2: **Yes:** Federico Cucci

Reviewer #3: **Yes:** Chipo Chimamise

---

## [Author Response · Author response to Decision Letter 1]

31 Dec 2025

University of Ghana

College of Health Sciences

School of Nursing and Midwifery

Department of Adult Health

26th DECEMBER, 2025

The Editor

PLOS ONE

Dear Sir/Madam,

Response to review

General comments of authors

We have addressed all the comments of the editor and reviewers as suggested and we hope the revised manuscript meets the standards for publications.

Comments of the Editor

PONE-D-25-57892

Health seeking behaviors of nurses diagnosed with hypertension and providing health care in resource-constrained setting in a rural part of Northern Ghana: A qualitative study

PLOS One

Dear Dr. Konlan,

Dear Dr. Konlan,

Thank you for submitting your manuscript to PLOS ONE. After careful consideration, we feel that it has merit but does not fully meet PLOS ONE’s publication criteria as it currently stands. Therefore, we invite you to submit a revised version of the manuscript that addresses the points raised during the review process.

• A letter that responds to each point raised by the academic editor and reviewer(s). You should upload this letter as a separate file labeled 'Response to Reviewers'.

We look forward to receiving your revised manuscript.

Kind regards,

Javier Fagundo-Rivera, PhD

Academic Editor

PLOS One

Journal Requirements:

2. We note that your Data Availability Statement is currently as follows: All relevant data are within the manuscript and in Supporting Information files.

Additional Editor Comments:

Dear Authors,

Thank you for your work and for your interest in PLOS ONE as a venue to disseminate your research.

Three reviewers have evaluated your manuscript. A major revision is required, addressing the comments from all three reviewers.

Please revise your manuscript accordingly and submit a detailed response-to-reviewers letter indicating how you have addressed each of the reviewers’ comments.

Kind regards,

Authors’ Response to Comments of editor

General Response to editor’s comments

We are grateful for the comments of the editor and have addressed all the concerns and comments of the editor.

Authors’ response to Journal Requirements:

1. We have ensured that our manuscript meets PLOS ONE's style requirements, including those for file naming.

2. We have ensured that our ethics statement appeared only in the Methods section of our manuscript. This is found on page 11 of the revised manuscript.

3. We have stated on page 11 that all the participants gave consent for interview transcript to be published.

4. We have removed all personal identification information in the revised manuscript

5. This was not applicable to our manuscript.

Additional Editor Comments:

We have addressed the comments of the editor in the revised manuscript.

REVIEWER 1 COMMENTS

Reviewer #1: SEE DOCUMENT ATTACHED

This study, titled "Health seeking behaviors of nurses diagnosed with hypertension and providing health care in resource-constrained setting in a rural part of Northern Ghana: A qualitative study," explores the personal health management strategies and associated challenges faced by nurses living with hypertension in a high-stress, resource-constrained environment in Ghana. The manuscript provides valuable insights into the persistent gap between high professional knowledge and poor health-seeking behavior, attributing this divergence primarily to organizational and systemic barriers. I reviewed the study following PLOS ONE criteria which I now submit.

Authors’ response to reviewer 1

We are grateful for the comments of the reviewer. We have addressed the grammatical errors and we are grateful for the favourable comments of the reviewer in the attached filed.

Reviewer 2 comments

Reviewer #2:

I recommend major revision: (1) sample/ID inconsistency (n=12 but quotes labeled “Nurse 13”); (2) clarify recruitment, voluntariness, and privacy (management involvement; interview location); (3) align the Data Availability statement with PLOS requirements; (4) correct the Limitations section.

Authors’ responses to comments of reviewer 2

1. We have ensured the sample/ID are consistent and removed the error of nurse 13 found in the manuscript. This is found on page 6 on the sample size and also in the results on pages 12 to 28 of the revised manuscript.

2. We have clarified the recruitment, voluntariness, and privacy (management involvement; interview location). This is found on page 8 of the revised manuscript on selection of participants and data collection.

3. We have aligned data availability with PLOS requirements as suggested by the reviewer.

4. We have worked on the limitations section of the revised manuscript. This is found on pages 31 and 32 of the revised manuscript.

Reviewer 3 comments

Reviewer #3: SEE DOCUMENT ATTACHED

The manuscript is scientifically sound and backed by data. The authors need to correct grammar mistakes and try not to mix discussion with results, otherwise the manuscript reflects good scientific rigor and the conclusions and recommendations tie up well with the findings of the study. With minor corrections, it is a good manuscript for publication.

Authors’ responses to comments of reviewer 3

We are grateful for the comments of the reviewer.

We have worked on the grammatical mistakes and ensured that the discussion is separated from the results in the revised manuscript.

We have addressed all the concerns of the reviewer.

We hope the revised manuscript will meet the standards for publication.

Thank you

Yours sincerely,

Dr. Kennedy Dodam Konlan

(Corresponding author)

---

## [Decision Letter · Decision Letter 1]

22 Jan 2026

Health seeking behaviors of nurses diagnosed with hypertension and providing health care in resource-constrained setting in a rural part of Northern Ghana: A qualitative study

PONE-D-25-57892R1

Dear Dr. Konlan,

We’re pleased to inform you that your manuscript has been judged scientifically suitable for publication and will be formally accepted for publication once it meets all outstanding technical requirements.

Kind regards,

Javier Fagundo-Rivera, PhD

Academic Editor

PLOS One

Additional Editor Comments (optional):

Dear Authors

Congratulations for your work. This manuscript can be accepted.

Kind regards.

Reviewers' comments:

Reviewer's Responses to Questions

**Comments to the Author**

Reviewer #1: All comments have been addressed

Reviewer #2: All comments have been addressed

Reviewer #3: All comments have been addressed

2. Is the manuscript technically sound, and do the data support the conclusions?

Reviewer #1: Yes

Reviewer #2: Yes

Reviewer #3: (No Response)

3. Has the statistical analysis been performed appropriately and rigorously?

Reviewer #1: Yes

Reviewer #2: Yes

Reviewer #3: (No Response)

4. Have the authors made all data underlying the findings in their manuscript fully available?

Reviewer #1: Yes

Reviewer #2: Yes

Reviewer #3: (No Response)

5. Is the manuscript presented in an intelligible fashion and written in standard English?

Reviewer #1: Yes

Reviewer #2: Yes

Reviewer #3: (No Response)

Reviewer #1: This revised manuscript demonstrates substantial improvement in clarity, structure, and methodological transparency. The authors have engaged seriously with previous reviewer comments and have strengthened the manuscript in ways that make it both analytically rigorous and highly relevant for publication. The study now presents a coherent, well-documented, and ethically sound contribution that is appropriate for the scope of PLOS ONE. Considering the revisions made and the overall quality of the manuscript, I recommend acceptance.

Reviewer #2: The manuscript “Health seeking behaviors of nurses diagnosed with hypertension and providing health care in a resource-constrained setting in a rural part of Northern Ghana: A qualitative study” is ready for publication.

Reviewer #3: (No Response)

**Do you want your identity to be public for this peer review?** For information about this choice, including consent withdrawal, please see our Privacy Policy

Reviewer #1: No

Reviewer #2: **Yes:** Federico Cucci

Reviewer #3: **Yes:** Dr Chipo Chimamise

---

## [Editor Report · Acceptance letter]

PONE-D-25-57892R1

PLOS One

Dear Dr. Konlan,

I'm pleased to inform you that your manuscript has been deemed suitable for publication in PLOS One. Congratulations! Your manuscript is now being handed over to our production team.

Kind regards,

on behalf of

Dr. Javier Fagundo-Rivera

Academic Editor

PLOS One